# Current Understanding of the Neutrophil Transcriptome in Health and Disease

**DOI:** 10.3390/cells10092406

**Published:** 2021-09-13

**Authors:** Luke W. Garratt

**Affiliations:** Wal-Yan Respiratory Research Centre, Telethon Kids Institute, University of Western Australia, Nedlands 6009, Australia; luke.garratt@telethonkids.org.au; Tel.: +618-6319-1804

**Keywords:** RNA sequencing, innate immune response, drug screening, granulocyte

## Abstract

Neutrophils are key cells of the innate immune system. It is now understood that this leukocyte population is diverse in both the basal composition and functional plasticity. Underlying this plasticity is a post-translational framework for rapidly achieving early activation states, but also a transcriptional capacity that is becoming increasingly recognized by immunologists. Growing interest in the contribution of neutrophils to health and disease has resulted in more efforts to describe their transcriptional activity. Whilst initial efforts focused predominantly on understanding the existing biology, investigations with advanced methods such as single cell RNA sequencing to understand interactions of the entire immune system are revealing higher flexibility in neutrophil transcription than previously thought possible and multiple transition states. It is now apparent that neutrophils utilise many forms of RNA in the regulation of their function. This review collates current knowledge on the nuclei structure and gene expression activity of human neutrophils across homeostasis and disease, before highlighting knowledge gaps that are research priority areas.

## 1. Introduction

Neutrophils must be able to adapt and rapidly respond to a diverse range of environments. To meet this challenge, they have a broad range of abilities. These include canonical functions of phagocytosis of bacteria and fungi, production of oxygen species, and directing the recruitment of other leukocytes. Neutrophils can also display more advanced capabilities, including the generation of neutrophil extracellular traps (NETs), suppression of T-cells and other adaptive immune cells, as well as antigen presentation (all reviewed in [1]). It has been long known that neutrophils feature a significantly smaller resting gene expression profile than other cell types such as peripheral blood mononuclear cells (PBMC) [2]. However, the increasing sensitivity of molecular techniques over the past 25 years have highlighted that transcriptomic activity does occur in neutrophils [3] and expands rapidly during activation to regulate neutrophil functions [4]. Now, the latest single cell analysis techniques are revealing transcriptomically distinct populations, even amongst mature peripheral neutrophils [5].

This new area of neutrophil research is interesting for drug development, as exaggerated and/or aberrant neutrophil activity can be detrimental to tissues. Neutrophils are often the prominent cell in diseases featuring chronic inflammation, including cystic fibrosis (CF) and rheumatoid arthritis (RA). Neutrophils can also feature in autoimmune diseases such as multiple sclerosis and neutrophil infiltration can be prognostic for poor outcomes in some cancers. Current therapies for neutrophils target either broad inhibition of the upstream inflammatory signals or specific downstream products of neutrophil activity. The quandary is that these approaches can diminish the beneficial contributions of neutrophils. So there remains a need for new compound classes that can modulate specific responses by neutrophils. As next generation sequencing technologies enable analyses of the global neutrophil decision making process, these offer an enticing path to the goal of therapies to modulate neutrophil function(s).

The application of next generation sequencing to understanding myeloid cell biology has been recently appraised [6] and current knowledge on the regulation of general neutrophil function by transcription factors has also been published [7]. To narrow the focus of the review, studies that specifically interrogated neutrophils were prioritized over studies utilising bulk approaches to tissues or leukocytes. After briefly covering neutrophil development, new updates on the transcriptional infrastructure of neutrophils will first be described. The review will then delve into what has been discovered regarding gene expression by neutrophils in different disease and biological scenarios. Finally, some priority research questions for future studies will be outlined, including the need for more data on modulation of neutrophil transcriptomic activity by therapeutic compounds.

## 2. Neutrophil Development

Haematopoiesis in the bone marrow is responsible for generating much of the total leukocyte pool. Haemopoietic stem cells give rise to common myeloid precursors that, dependent upon differential induction of transcription factors C/EBP, GATA-1 and PU.1, can mature into monocytes or neutrophils (and to a far lower extent, eosinophils) [8]. Since neutrophils only have a half-life of 13–19 h in blood, to ensure that a pool of approximately 6.1 × 10^8^ neutrophils per kilogram of body weight is continuously maintained for immunocompetence, approximately 60% of bone marrow cells are dedicated granulocyte precursors [9]. For further detail, a combined transcriptome–proteome analysis of myeloid differentiation has been recently published [10] and a number of thorough reviews on the life cycle of mammalian neutrophils are available [1,11].

Concerted analyses of mature neutrophils in blood and tissues by flow cytometry has led to reports of a diverse array of subsets or activation states, typically based upon variations in the expression of surface markers or other proteins [12]. The two most prominent markers of subsets that can be reliably observed are CD177 (human neutrophil antigen NB1) [13] and olfactomedin 4 [14]. Further subset markers include CD10 [15], or variations of canonical markers, such as CD16, CXCR4/CD184, CD63, CD62L and CD54 [12,16]. It is still unclear for many subsets if they are bona fide lineages or simply differentiations and/or maturation states induced by the tissue environment(s). Certainly upon chronic inflammation there can be substantial increases in novel marker expression, which has been long known in respiratory diseases [17], as well as shifts in the maturity of systemic neutrophil populations during severe inflammatory trauma resulting in CD16^low^ and CD62^low^ populations [18]. Neutrophils have also been observed as rare events in mononuclear separations of blood and these cells, termed low density neutrophils, have been implicated in a variety of diseases [12]. Again, many questions remain over their definition and even what degree of functional exceptionality these low density neutrophils have from the rest of the neutrophil population [19]. Indeed, ‘naming’ subsets creates rules and restrictions that make understanding the true biology of a cell difficult. A new, unbiased approach to characterizing neutrophil heterogeneity is underway, excellently reviewed by Ng and colleagues [20]. For this review, neutrophil subsets will only be discussed if relevant to gene expression findings or transcriptional heterogeneity.

## 3. Neutrophil Transcriptional Infrastructure

### 3.1. The Distinctive Neutrophil Nucleus

The nucleus is the core of cellular decision making, housing DNA/RNA synthesis machinery and a homing target for transcription factors. Upon visual inspection of human blood smears, the neutrophil is immediately noted as having a uniquely structured, two to five-lobed nucleus. As well as proving a convenient marker in haematology for neutrophil identification and characterization, the assumption has been that this segmented nucleus is what gives neutrophils such prodigious migratory capacity. However, in recent years it has been established that the neutrophil nuclear envelope itself is different to other cell types, starting with higher amounts of lamin B receptors and lower amounts of the linker of nucleoskeleton and cytoskeleton (LINC) complex [21]. The neutrophil nuclear envelope also features uniquely low lamin A/C expression [22]. It is principally this feature, not the multi-lobed structure, that permits significant deformability of the neutrophil nucleus for fast migration through tight 1 µm gaps in tissues, which is too narrow for other cell types [22].

Reinforcing this hypothesis is the fact that these unique compositional characteristics not only become established as granulocytes terminally differentiate, they are highly conserved across species [23]. Neutrophils function normally with under-segmented nuclei, as cells from people with Pelger–Huët anomaly feature an ovoid nucleus but display unaffected migration, phagocytosis and respiratory burst [24,25]. Indeed many species feature neutrophils with round or ovoid nuclei [23]. The reverse can occur as well, with hyper-segmentation (>5 lobes) observed during granulocyte-colony stimulating factor (G-CSF) therapy [26]. Hyper-segmentation can also occur in diseased samples and certain in vitro conditions [27,28,29]. While the relationship of hyper-segmentation to neutrophil function has not been fully investigated, observing unaffected chemotaxis across the spectrum of neutrophil nuclei segmentation suggests that it did not evolve specifically to benefit migration.

### 3.2. Transcription and Nuclei Structure Intertwine

There is accumulating evidence that the structure of the neutrophil nucleus is instead linked to the requirements for transcriptional control following granulopoiesis and maturation. Certainly, it is becoming accepted that the low basal transcriptional activity by neutrophils is not a consequence of the segmented structure preventing chromosomal organisation, another long held assumption of neutrophil biology. Although the distribution of chromosomes to the neutrophils lobes is random [30], distribution within lobes is non-random and radial based upon chromosome size as seen in regular spherical nuclei [30]. Super-resolution microscopy has now shown that chromatin in neutrophils is extremely compacted at the lobe peripheries, in contrast to the loosely arranged networks seen in precursor cells and somatic cells with spherical nuclei [31]. Monocytes, cells with non-spherical horseshoe-shaped nuclei and also reduced transcriptional activity compared to other leukocytes, exhibit an intermediate structure with compacted chromatin islets [31]. Hübner and colleagues have also reported a relative lack of active RNA Pol II signals and tiny nucleoli in both neutrophils and monocytes (although more pronounced in neutrophils), aligning with the muted transcriptional activity in these cells [4]. Ribosomal DNA in neutrophils is also tightly sequestered to the nuclei lamina and results in a virtual cessation of ribosomal RNA synthesis [32]. Overall, genomes of mature neutrophils are characterized by extensive loss of local genomic interactions and super contraction into inactive heterochromatin, but enriched for inter-chromosomal interactions [32] to enable rapid induction of some genes (i.e., *CXCL8)* and not others (i.e., *IL-10*) [33].

In fact, the need for low basal transcriptional activity to freeze neutrophil granule production upon maturation may be the cause of the distinctive nuclei structure. The critical developmental process of granulopoiesis requires downregulation of multiple, specific genes as each different family of granules are formed and packaged prior to terminal differentiation [34]. Here lies a biological clue to the structure of the neutrophil nucleus. To downregulate genes for granule contents in maturing neutrophils, the housekeeping process of intron retention alternately splices these genes to induce mRNA decay [35]. Yet also subject to intron retention is lamin B, the major lamin in the neutrophil nuclear envelope. When Wong and colleagues generated a mouse with an intronless lamin B gene (*LMNB1*) that is unaffected by intron retention, both granulopoiesis and nuclei morphology were aberrant [35]. This adds to earlier findings that mutations in the laminin B receptor gene (*LBR*) result in the loss of nucleus lobulation [36] and that neutrophil genomes do not differ from those of mononuclear cells in the number of topologically associating domains [32]. A more recent study has demonstrated that the morphology of the neutrophil nucleus is indeed independent of the whole cell cytoskeleton [37], suggesting the morphology of neutrophil nuclei is likely determined by nuclear processes.

Furthermore, granulopoiesis and nuclei segmentation occur in parallel. It has been observed that many mutations in the gene for the primary granule protein neutrophil elastase (*ELANE*) cause neutropenia [38]. There are currently several hypotheses on the mechanism involved, most related to misfolded protein. However, it is notable that whole deletions of *ELANE* do not cause neutropenia like single-point mutations [39,40], hinting at a role for intron retention. Whilst more consistent data on intron retention in neutrophil maturation across the animal kingdom is needed, it does appear that nuclear segmentation is a beneficial consequence of a carefully regulated maturation process, focused on creating and then pausing the specialized library of granules needed for the frontline of innate immunity. This developmental strategy ensures that this major immune effector cell is quiescent until it encounters the right stimulation(s), where it rapidly springs into action using the many pre-packaged components ready within the cell. At this point, the transcriptional activity is also rapidly revived to assist neutrophils in appropriately responding to the complex environments they encounter. Here, the transcription factor PU.1 is again critical to neutrophil function. PU.1 deletion from mature neutrophils results in uncontrolled activity [41], indicating PU.1 binding helps to restrain the neutrophil epigenome and ensuring that neutrophil activation occurs in a coordinated manner. A very recent follow up study has uncovered that PU.1 binding can differ between individuals, due to genetic variation in up to 27 associated genes [42]. Most significantly, the extent of variation and reduced PU.1 binding was associated with increased likelihood of autoimmune diseases, such as inflammatory bowel disease, rheumatoid arthritis and ulcerative colitis [42].

### 3.3. Homeostatic Transcription

Mature neutrophils exhibit transcriptional activity during both homeostatic circulation and during canonical functions, summarised in Table 1. It is now clear that there is a diurnal pattern to the characteristics of blood neutrophils due to an internal timer regulator [43]. Expression of multiple adhesion markers, including CD11b, ICAM-1, CD62L and CXCR4 [43], as well as variations in functional capacities, such as reactive oxygen species (ROS) generation, all fluctuate in a diurnal manner [44]. While the activity of the neutrophil transcriptome is low, diurnal variations can be measured, predominantly in genes involved in the Toll-like receptor and CXCR2 signalling, adhesion and cell death pathways [43,45]. Analyses of hundreds of donors has also uncovered that blood neutrophils actually feature greater transcriptional and epigenetic variation than CD14+ monocytes and CD4+ T cells [46,47], which likely enables the broad functional plasticity required for their continuously dynamic homeostasis.

The transcriptomes of mature blood neutrophils can be altered by even minor environmental cues, with gene expression changes observed after brief bouts of exercise [48]. A substantial number of these genes were unique to neutrophils, with only 16% overlap to PBMC gene responses to exercise. The follow-up study by the authors revealed that alterations in neutrophil expression of 38 microRNA (miRNA) drove a quarter of these exercise responses genes [56], demonstrating that this important class of RNA regulators play an important role in neutrophil function during normal physiology. In addition to miRNAs, time dependent expression of long noncoding RNA (lncRNA) also govern function of neutrophils and other immune cells [57]. One critical lncRNA is *Morrbid*, which regulates the transcription of the neighbouring pro-apoptotic gene, *Bcl2l11,* to control the myeloid cell lifespan [58]. The role of lncRNA to work in conjunction with mRNA gene expression in prompting spontaneous, constitutive apoptosis has also been recently investigated [59].

Since neutrophils must nearly always migrate from the peripheral blood through the endothelium and into tissue, understanding how migration impacts the neutrophil transcriptome is also important. Indeed some neutrophil responses, such as degranulation, only occur or are dramatically elevated if the cells are undergoing migration [60,61]. Jacobsen and colleagues compared the effects of migration through large (14 µm) and constricted (5 µm) pores on neutrophil chromatin and gene expression [49]. Common to both migrations were 420 genes with roles in expected biological processes of response to cytokine and monocyte chemotaxis. Migration through the constricted space led to four-fold more gene expression (2041 genes) over unmigrated cells in comparison to the large pore migration (520 genes differentially expressed from unmigrated). Within this were 199 genes unique to the constricted migration. These 199 genes were associated with biological processes of actin cytoskeleton remodelling, lactate transport and Rab GTPase activity [49]. Gene expression occurred with minimal compartment switching of chromatin and there was no correlation of genes with region stability. This study further highlights the unique genome structure of neutrophils to provide extreme mechanical flexibility whilst retaining capacity for transcriptional regulation.

Evidence is also building for a significant role of neutrophils in healthy tissue homeostasis beyond their canonical function. Firstly, it has been effectively illustrated how the fate of neutrophils are determined by the tissues, that co-opt these cells for tissue maintenance [62]. Analysis by scRNAseq and mass cytometry highlighted distinct difference in gene expression and surface protein profiles between bone marrow, blood, kidney, lungs, intestine and skin [62]. Once they have departed the blood, these neutrophils have been reported to substantially influence the transcriptomic profiles of other leukocytes and tissue cells themselves [62]. Neutrophils also have an important role in regulating wound repair. Excess persistence or activity of neutrophils at wound sites results in non-healing wounds and can be observed in disease, such as diabetes, where neutrophils are primed to undergo NETosis [63]. In a related study it was discovered that neutrophils express mRNA for *F12*, the gene encoding coagulation factor XII. In mouse models of chronic sterile inflammation, deletion of *F12* resulted in reduced neutrophil numbers and NETosis and faster wound healing, highlighting how important regulation of the neutrophil response is to wound resolution. Finally, a novel homeostatic function for neutrophils in the marginal zone of the spleen has been reported [64], although this has been recently disputed [65]. In the spleen, neutrophils develop a transcriptional profile that provides them with unexpected B-cell stimulating properties, working in conjunction with splenocytes to switch immunoglobulin production by B cells [64]. Disruptions to this process have been proposed in multiple sclerosis, where mouse models reveal neutrophils precede influx of B cells into the central nervous system and become involved in B cell recruitment [66]. Neutrophils from people with multiple sclerosis exhibit increased responsiveness to stimulation over non-disease neutrophils [67]. It is also noted that B cells in the marginal zone express high levels of the receptor for B cell activation factor (BAFF) and splenic neutrophils express the highest levels of the *Baff* gene amongst immune cells [68].

## 4. Transcriptional Responses during Disease

The literature on transcriptional activity in neutrophils across different disease scenarios is continuing to grow. The following sections review this literature in detail, but Table 2 summarises some key transcription molecules that have been reported to date.

### 4.1. Infection Response

It is now 15–20 years since initial studies have demonstrated that neutrophils are capable of reviving gene expression in response to classical infectious stimuli [83,84,85,86]. These reports are perhaps unsurprising considering the already recognized ability of neutrophils to generate and release cytokines de novo [2]. Investigations into canonical neutrophil activation (migration and phagocytosis) regularly observe involvement of the NF-κB subunits and STAT3 proteins in neutrophil activation [7]. NETosis, the process of NET formation, is also transcriptionally dependent, as demonstrated by the ability of the global transcription inhibitor actinomycin D to almost completely suppress NETosis [54]. Key to regulating the formation of NETs from this transcriptional activity are the miRNAs miR-155, miR-146a, miR-505 and miR-378a-3p [87]. These can be from endogenous miRNAs or exogenous miRNAs from exosomes of macrophage and tissue cells, or activated platelets [87].

Investigations of neutrophil responses to infection have recently been advanced into the modern realm of single cell RNA sequencing [5]. The first major advance by this study was that three transcriptionally distinct neutrophil subsets could be identified in the mature neutrophil population in peripheral blood, which were common across humans and mice. There was a canonical neutrophil population expressing *MMP8*, *S100A8* and inflammatory response genes, as well as an “aged” population based upon the high relative expression of *CXCR4* [5] that fits with accepted convention of subsets observed by flow cytometry [12]. The third population featured an interferon dominant transcriptome of *IFIT1* and *ISG15*, and the authors presented a flow cytometry protocol to isolate this IFIT1^+^ population. When mice were challenged with intraperitoneal *E. coli* infection, the transcriptional regulatory networks underwent a consistent drift to prioritize cellular resources towards defense responses, and neutrophil maturation was accelerated whilst the proportion of the interferon dominant subset doubled [5]. However, within this framework, all three subsets exhibited some unique gene ontologies in response to infection. This discovery has major implications for how research into infectious diseases assesses the neutrophil population in blood and tissue and to understand whether poor bactericidal activity is from defective mechanisms, population switching, or both.

Other studies have looked beyond the classical workhorse bacteria *E. coli* and assessed neutrophil transcriptomic responses to more clinically interesting pathogens. Gomez and colleagues compared lung neutrophils from mice with *Streptococcus pneumoniae* lung infection versus mice challenged with the vehicle control [70]. Neutrophils migrating into *S. pneumoniae* lungs featured 4127 differentially expressed genes over the vehicle treated lungs. As expected, gene set enrichment analysis revealed the top 25 enriched pathways covered response pathways to bacterial molecules, innate immune response and cytokine production [70]. However, pathways related to regulation of adaptive response were also significantly enriched, supporting a role for lung neutropohils in directing the adaptive immune cells. Infections with another cause of the pneumonia, *Yersinia pestis*, which was responsible for the pneumonic plague, result in massive neutrophilic infiltration into the lung. A study compared central, neutrophil dominated lesions from mouse lungs infected with *Y. pestis* versus the peripheral lung areas by spatial RNAseq, both cross referenced with bone marrow neutrophils [69]. This analysis yielded a 224 gene signature in central lesion neutrophils that was dominated by downregulation of genes in apoptosis pathways and leukocyte migration [69]. An interesting observation to this infection model was that despite residing in a highly inflammatory microenvironment, the transcriptome of neutrophils in the infected lesion centre was more like unstimulated bone marrow neutrophils than the actively migrating neutrophils in the peripheral areas of the lung. Further investigation in this study revealed that *Y. pestis* extends neutrophil survival to its benefit, reducing bacterial clearance and permitting lesion formation [69]. This study highlights how embracing a diverse range of bacterial microbes can be greatly beneficial in elucidating mechanisms of neutrophil function, as well as the ever-adapting challenges faced by the innate immune system. A more recent study performed scRNAseq on amniotic fluid that were culture positive for a variety of bacteria [88]. Neutrophil and monocyte/macrophage cells were isolated by FACS and DNA fingerprinting was also applied to determine the maternal or foetal origins of the myeloid cells. The authors demonstrated that the neutrophil transcriptome only differed slightly between maternal or foetal origin samples, with 66 genes downregulated and 53 upregulated in foetal neutrophils and no specific enrichment of biological pathways [88]. Differences in gene expression between neutrophils from term versus preterm birth cases were of a similar number. The authors did not provide details on the genes differentially expressed. Overall, it suggests that even in foetal development, neutrophils are already transcriptionally regulated in a similar manner to adult neutrophils.

Fungal organisms are another infectious risk to humans. Approximately 50% of the fungi sensing receptors are expressed on neutrophils to facilitate anti-fungal immunity. An early study of the neutrophil response to *Candida albicans* measured changes in 191 genes, of which 65% were upregulated [89]. It was noted that granule proteins did not feature in the upregulated genes and the authors observed that neither inhibition of RNA or protein synthesis affected the immediate capacity to kill *C. albicans* [89]. Ten years later, a RNAseq study expanded this to 318 genes [53] and usefully, also performed analyses across three time periods post-infection. Neutrophils differentially regulated less than 40 genes during the initial 30 min of encountering *C. albicans* yeast or hyphae, but by 60 min this has expanded to more than 250 genes [53]. A key gene was *NLRP3*, which is the only NOD-like receptor to detect fungi and the authors demonstrated that *C. albicans* stimulated inflammasome activation in neutrophils.

Many viruses can result in neutrophilic infiltration, particularly the seasonal respiratory viruses of influenza, respiratory syncytial virus and rhinovirus (recently reviewed [90]). There is currently a lack of studies that have assessed neutrophil transcriptomes during viral infection. The first studies have focused on influenza A virus as neutrophils express sialic acid receptors and can be infected [91,92]. One study analysed gene expression of a mouse neutrophil progenitor cell line and demonstrated that neutrophils can recognise the influenza virus particle by internal pattern recognition receptors [71]. This resulted in increased transcription within 3 h, generating a core type I interferon response (including *IFNβ1*, *ISG15* and *IRF1*), antiviral proteins (*CXCL10*), along with apoptosis pathway genes (*CASP4*, *NOD1* and *PARP14*) [71]. Observations from another study using a mouse model of influenza infection reported that expression of transcriptional regulator B cell lymphoma 6 (BCL6) is upregulated in neutrophils that have migrated to the influenza affected lung, possibly to repress neutrophil apoptosis [93].

The global COVID-19 viral pandemic has prompted a strong revisiting of neutrophil responses to viruses, as infection by the SARS-CoV-2 virus can result in severe lung inflammation and cytokine storms [94]. Early in the pandemic, existing genomic datasets were leveraged in a study that proposed a link between SARS-CoV-2 and neutrophil degranulation [95]. Unique to COVID-19 has been the rapid implementation of scRNAseq to peripheral blood and bronchoalveolar lavage samples, with the first data published within months [96]. This publication and following studies [72,73,97,98,99] or re-analyses [100,101] are providing a rich resource of data on neutrophil transcriptomes in COVID-19 across varying severities of viral load and respiratory distress. The current state of these findings is that neutrophils are abundantly recruited to airways in severe but not mild disease, and their gene expression signatures strongly featured pro-inflammatory cytokines (*CXCL8)* as well as *CTSB, CSTD, HIF1A* and *CYBB*, genes involved in NETosis pathway [100]. Analysis of peripheral blood samples by scRNAseq indicates that this abundant recruitment and cytokine signalling results in severe pressure on granulocyte production, with emergency myelopoiesis leading to increasing presence of immature and dysfunctional neutrophils [72]. Another study performed bulk-RNAseq on isolated granulocytes and the gene expression data could discriminate mild from severe patients [102]. Both this data and whole blood bulk-RNAseq data revealed a strong enrichment of neutrophil activation signatures in severe disease, with heightened expression of *CD177* and *S100A12* in the first 10 days [102]. Together, these ongoing studies are identifying similar trends of activated mature neutrophils in mild disease and highly activated, NET forming immature neutrophils in severe disease. This overlaps with Kawasaki syndrome, which is being observed particularly in children and a disease where neutrophil gene activity has also been recently investigated [103].

### 4.2. Chronic Inflammation

In addition to acute response, the transcriptional activity of neutrophils is being interrogated in many chronic inflammatory diseases. An early study compared blood neutrophils from healthy children, children with juvenile arthritis and children with CF [75]. The authors were able to establish a signature of 148 genes that were common to blood neutrophils during soft tissue inflammation, and just 68 genes were unique to juvenile arthritis. More importantly, the authors investigated miRNAs in these samples and observed specificity between juvenile arthritis and CF, both in the miRNAs expressed and the gene isoforms present. Complex regulatory networks were revealed and the hubs of the miRNA networks were distinct to each disease phenotype, highlighting the ability of neutrophils to make subtle programming alterations. Later RNAseq work in CF blood neutrophils by this group also revealed that gene expression by peripheral blood neutrophils also varied during exacerbations, with 136 transcripts changing from convalescence [104]. Three-quarters of genes were upregulated, with the proposed neutrophil subset marker, *CD177,* the most upregulated gene, but also featuring increased expression of inflammasome pathway genes *AIM2* and *NLRP3* along with *S100A12* [104]. Intriguingly, another subset marker, *OLFM4*, was the most downregulated gene during exacerbation, along with genes for other granule associated proteins including *MPO*, *PRTN3* and *ELANE*. The authors also found lncRNA were commonly expressed during exacerbation, with 34 expressed in at least one quarter of the subjects, further demonstrating the breadth of neutrophil transcription potential in peripheral blood.

The changes in the peripheral neutrophil transcriptome that occur during antineutrophil cytoplasmic antibody-associated vasculitis were recently described as able to distinguish not only disease from healthy controls, but also active disease from remission [76]. Hub genes in the neutrophil gene module featured the proinflammatory molecules *S100A12* and *S100A9* and the module was also highly enriched for NETosis related genes, including the central NETosis regulator *PADI4* [76]. Transcriptional activity of neutrophils may also be repressed in certain chronic diseased environments. This has been shown indirectly in a study of endotoxemia in early atherosclerosis, which observed that neutrophils were polarized into a non-resolving inflammatory state, with the homeostatic transcription factors *ATF4* and *KLF2* repressed [105]. In rheumatoid arthritis, repression of interferon gene networks in neutrophils was associated with poor response to tumor necrosis factor inhibitors [106].

Neutrophils change substantially upon migration into the inflamed lung tissue [1]. The lung can be affected by multiple disorders of chronic inflammation, with CF perhaps the most well studied. Neutrophils migrating into the CF airway develop a high degranulation phenotype [107,108] that underlies the high protease burden driving lung damage [109]. Follow-up work by the Tirouvanziam group analysed the transcriptional activity of separated neutrophils migrating into a model of the CF airway environment [61]. A median 3.5-fold increase in RNA content occurred over the 10 h, concurring with in vivo observations. This led to 6750 genes being differentially regulated from blood neutrophils, with up and down regulation occurring relatively equally [61]. Active migration of neutrophils was a critical requirement for this degree of transcriptional activity, demonstrating the importance of modelling the complete neutrophil experience from when it departs circulation into the tissue. Furthermore, Margaroli and colleagues could prevent neutrophils forming the high degranulation phenotype upon migration by blocking de novo transcription with α-amantin. Studies are now applying scRNAseq to lung samples and although filtering of low-level transcripts or use of cryopreserved samples can be common, both of which impact the resolution of neutrophils in these datasets, there is a growing database interrogating neutrophil transcription in healthy and disease lungs [69,74,81,82,110,111]. One recent scRNAseq study investigated sputum from adults with CF and age matched health controls [82]. As per the study by Xie and colleagues [5], this CF sputum study also reported three neutrophil populations. However, these were distinguished by canonical immature (*CXCR4* and *IGF2R*) and mature neutrophil (*FCGR3B*, *ALPL*, *CXCR2*) markers and heat shock proteins. This study also highlighted that many counterproductive signals could be observed across the complex spectrum of airway neutrophils, demonstrating the strength of scRNAseq over bulk-seq approaches in generating clearer understanding of pathological transcription states. Other non-COVID-19 pulmonary scRNAseq studies have either not analysed neutrophils in their dataset [111], or did not report discovery of transcriptomically distinct sub-populations of neutrophils [110]. The latter study reported pulmonary neutrophils could display high expression of the *Retnlg* gene [110], significantly elevated over the transcription levels within blood neutrophils. Resistin levels correlate to negative lung function in CF [112].

### 4.3. Cancer

Neutrophils are known to have antitumor potential but may also be prognostic for worse outcomes, so there have been efforts to understand the underlying biology of neutrophils in tumors. This section will focus on neutrophil gene expression in cancer as there is an excellent recent review of neutrophil heterogeneity in cancer [113]. Multiple models of tumor immunology have shown that several cytokines in tumors induce changes in neutrophil gene expression. The original studies of neutrophils in cancer identified a pro-tumorigenic ‘N2′ neutrophil phenotype, with this polarisation driven by lack of type-1 interferons and presence of TGFβ [77]. Gene markers for N2 neutrophils include *CD206*, *CCL2* and *ARG2* and they are transcriptomically distinct from both their tumor inhibitory ‘N1′ counterpart, and granulocyte derived myeloid-derived suppressor cells (G-MDSC) [78]. Furthermore, N2 polarisation can be induced by nicotine exposure [114].

To understand the polarization process, many groups have investigated neutrophils following exposure to cytokines in cancer pathology. In response to TGFβ, N2 neutrophils were noted for their downregulation of several immune response pathways, notably in antigen processing and presentation (*CALR*, *LGMN*), chemokine release (*CXCL13*, *CXCL10* and *TNFα*) [79]. Other studies have investigated IL-6 and G-CSF, as well as IL-35, an inhibitory cytokine of many immune cell types that limits anti-tumor immunity [115]. These upregulate the expression of *MMP-9* and *Bv8*, proteins involved in neutrophil promoted angiogenesis, and downregulate the angiogenesis suppressing *TRAIL* [116]. Furthermore, neutrophils from IL-35 over-expressing mice suppressed T-cell proliferation through inducible nitric oxide synthase (iNOS), with these neutrophils featuring upregulated *Nos2* expression [116]. Another study investigating development of liver metastases also analysed transcriptomic responses by neutrophils to cytokines. Mouse bone marrow neutrophils were stimulated with G-CSF, as well as lipopolysaccharide, interferon γ and IL-4 [117] and analysed by RNAseq. Wang and colleagues only reported differential expression of 19 genes for purinergic receptors dependent upon the stimulating cytokine [117]. However, this study and others provide a trove of extensive data that could be mined, to understand whether neutrophil intracellular signalling to these factors is similar or different from other cell types. As well as cytokine gradients, neutrophils encounter more complex signals in the form of extracellular vesicles. It was recently demonstrated that extracellular vesicles from human gastric cancer induce PD-L1 gene and protein expression in neutrophils, generating an immunosuppressive phenotype [80]. Whilst this study did not assess the full transcriptome, it reflects a growing acceptance of analysing gene expression to understand neutrophil response to stimuli. Collation of the publicly available data on neutrophil gene expression to multiple host and external stimuli would provide a valuable resource for interrogating how neutrophils are influenced by the diverse environments they encounter.

Direct analyses of tumors or the accompanying splenic neutrophil populations by scRNAseq are now expanding the N1/N2 definition into a continuum of neutrophil states [81,118,119]. These states appear to be relatively consistent across species and could be resolved into as many as five to six subsets in tumors [81,118]. Zilionis and colleagues applied scRNAseq to non-small cell lung tumors in humans as well as mouse models. From their data, it appears that neutrophils proceed from a canonical precursor population through two to three intervening states to form the polarized N2 population (referred to as /’N_5’_^’^ by Zilionis and colleagues [81]). Stemming from a canonical population expressing *S100A8, S100A9*, *MMP9* and *ADAM8*, the tumor specific subsets were characterized by *CASS4*, *CTSC,* or *CCL3* [81]. The authors also reported a continuum of neutrophil states in peripheral blood of patients with cancer, although this was not directly analogous to the neutrophil continuum observed in the tumor. High-level analysis of the total neutrophil population revealed that 719 genes that were differentially express from their blood counterparts [81]. This study is also notable for identifying the interferon dominant, *IFIT1* neutrophil subset in tumors and blood, that has been reported across scRNAseq studies investigated in detail by Xie and colleagues [5]. Indeed, Zilionis and colleagues observed the neutrophil continuum in tumor and blood was separate to the interferon dominant population. This fits well with the observations of N2 neutrophils developing in areas of low type I interferons and may provide a clue to targeting the expansion of this interferon population in people with cancer. One important note regarding these subsets is that unlike those seen for monocyte and macrophages, whose subsets could be consistently detected in low numbers of patients, delineation of all neutrophil subsets required data from the majority (6/7) of patient samples [81]. It is likely that as the number of scRNAseq studies increase and the disease scenarios investigated become more comprehensive, the true continuum of neutrophil states will be better resolved.

Neutrophils may also have a role in determining cancer metastases [117]. Wang and colleagues observed that in pancreatic cancer a neutrophil subset, which did not express the purinergic receptor P2XR1 and were immunosuppressive to T cells, accumulated in livers with metastases [117]. This study utilised bulk RNAseq and so it is unclear how this subset relates to neutrophil subsets identified through scRNAseq, with P2XR1 not reported to date in single cell studies [81,118]. The transcription factor *Nrf2* was found to be elevated in P2XR1^-ve^ neutrophils and essential to the immunosuppressive function. Predominantly increased in P2XR1^-ve^ neutrophils were gene expression networks for metabolic pathways and the authors demonstrated that this was accompanied by elevated metabolic function and oxidative capacity.

## 5. Priority Research Areas

Over the last 50 years, advancing technology has changed our focus when studying neutrophils, shifting from mainly functional comparisons through to protein differences and now delving deep into the gene level (Figure 1). Accordingly, there exists a greater emphasis to routinely characterise the transcriptome as part of any study into neutrophil behaviour in health and disease. To date no clinical trials have stemmed from outcomes of neutrophil transcriptome studies, although a small number of new trials are now incorporating transcriptomic analyses as additional exploratory outcomes. By applying system biology techniques to elucidate the upstream networks and receptor signalling governing neutrophil fate(s), these analyses of the neutrophil transcriptome should identify therapeutic interventions that can modulate these networks, as is occurring in other cell types [120,121]. The difficult goal will be to dampen harmful neutrophil activities, without disrupting their important canonical functions. Achieving this goal will benefit from more applications of scRNAseq technologies to highly purified neutrophil populations, so as to reveal the transcriptional profiles that underly their functions across tissues and whether individual subsets can be therapeutically targeted. Employing multiple timepoints in these studies should also be considered, to provide insights into what aspects of neutrophil biology are executed by post-translational events and which require de novo transcription, and how they may be preferred for therapeutic intervention. Efforts to standardize neutrophil population nomenclature [1,12,20] should also be continued to help with reproducibility in an ever increasing complex field. Agreed protocols for the critical steps upstream of transcriptomic analysis would also benefit data sharing and reproducibility. It has been reported that isolation and preparation techniques impact neutrophil gene expression studies more through reducing yields and do not largely affect the overall gene expression profile [122]. Finally, systematic approaches to identify and characterise neutrophil subsets by high parameter flow cytometry are needed to validate causal relationships to their function. This last aspect is essential whilst the costs for scRNAseq remain prohibitive.

Unsurprisingly for a relatively nascent field, there is little literature on how different classes of compounds impact the neutrophil transcriptional framework, with publications largely describing effects on downstream functions. To the author’s knowledge, there are no studies assessing neutrophil transcriptomes in response to currently available therapeutics. Most studies have predominantly investigated cytokines that may have a therapeutic use, for example, stimulation of neutrophils with G-CSF, TNFα or IFN-γ, which resulted in distinctive transcriptional networks over unstimulated cells [50,117]. To realize the full potential of how systems biology can improve neutrophil therapies in disease, more data is needed on how the transcriptional networks of neutrophils are changed by common therapeutics/compound families and whether these occur similarly or differentially to other cells.

Finally, studies to date have usually employed a broad inhibition of *de novo* neutrophil transcription to demonstrate that the observed neutrophil functions are dependent on gene expression. Cellular transfection is a more refined technique in cell biology to manipulate specific gene expression. However, transfection of neutrophils can be notoriously difficult due to their short lifespan preventing lengthy protocols. While successful transfection has been reported in primary human neutrophils with varying efficiency [123,124], the neutrophil biology field would also benefit immensely from protocols to reliably achieve rapid and effective transfection of mRNA or siRNA into primary human neutrophils, prior to experimental manipulation and avoiding the use of imperfect cell line representatives, such as HL-60 cells.

## 6. Conclusions

Over the past 20 years, yet another dogmatic assumption in biology has been overturned, with early discoveries of neutrophil gene expression upon activation evolving into today’s nascent field of neutrophil systems biology. However, the issue remains that the most numerous initial responding cell type is often left out of clinical research investigations. Reasons include long-standing preference for isolating and working with mononuclear cell populations and poor awareness of the advances made in neutrophil biology. With the current state of knowledge on how neutrophils can influence many different cell types and the capability of modern technologies to analyse their complex biology in granular detail, there remains no reason for not embracing neutrophil transcriptomics in solving today’s medical challenges.

## Figures and Tables

**Figure 1 cells-10-02406-f001:**
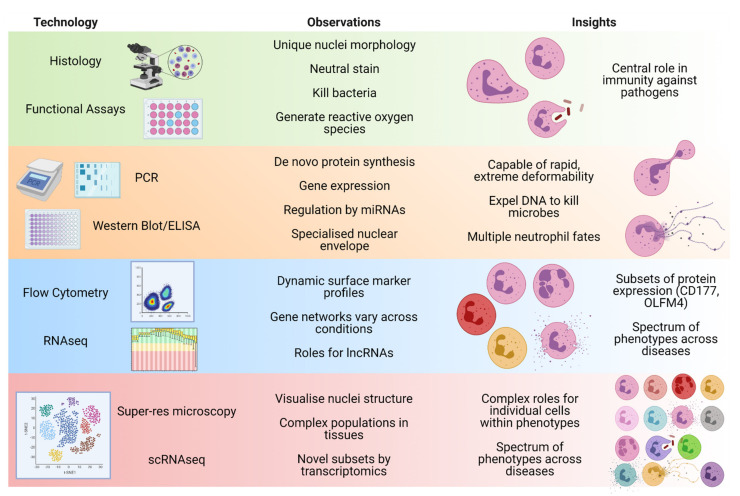
Summary of the observations of neutrophil features and insights into neutrophil biology made as technologies have advanced. Created with BioRender.com, 16 July 2021.

**Table 1 cells-10-02406-t001:** Summary of upregulated gene expression during neutrophil homeostasis and major functions. Genes featured are restricted to those highlighted by the original authors and/or the largest fold change reported by the study. Bold indicates common to the studies cited, underlined indicates common to multiple biological scenarios. RNAseq—RNA sequencing; NETosis—neutrophil extracellular trap formation.

Biological Scenario	Analysis Methods	Neutrophil Gene Upregulation Observed	Citations
Diurnal Homeostasis	RNAseq	Fluctuations in *Cry2*, *Arntl*, *Clock*, ***Per1***, *Cxcr5*, *Vav2*, *Cxcl2*, *Icam1*, ***Cxcr4***, *Sell*, *Cxcr2*, *Tlr4*, *IL17A*, *Csf3r*, *Mcl1*, *Fgr*, *Il1b*, *Il13ra1*, *Lmnb1*, *Atm*, *Dbp*, *Rev-erbα*, *CYBB*	[43,44]
Aerobic Exercise	Microarray	*GZMA, GZMH, PRF1, GZMB, HSPA1B, AREG, FYN, PDGFD, KSP37, SPON2, STAT4, TGFBR3, LDLR, SOCS2, RUNX3, KLRC4, ZAP70, SLAMf7, CTSW, GNLY*	[48]
Small pore migration	RNAseq	*EGR3, SERPINE1, CCL3, CCL4, EGR2, EGR1, TNF, FOSB, CCL4L2, CXCL8, BCL3, IER3, NFKBIA, TREM1, FOS, BTG2, NAB2, NR4A2, NFKBID, DUSP2, CD69*	[49]
Chemoattractant Priming (TNFα/GM-CSF/fMLF/LPS stimulation)	Microarray, RNAseq	** * CCL3 * ** *, **CCL4**, **CD69**, CISH, CXCL1, CXCL2, DUSP2, EDN1, EGR1, **EGR2**, GADD45B, GPR84, HBEGF, HCAR2, HCAR3, HRH4, ICAM1, IL1A, **IL1B**, IL1RN, KCNJ2, MFSD2A, NFKBIA, **NFKBIE**, OLR1, PDE4B, PLAU, PNPLA1, PPP1R15A, **RHOH**, SLC35B2, SOCS3, TARP, TIFA, TNF, TNFAIP3, TNFAIP6, TRAF1, ZFP36, BIRC3, CCR1, IL3RA, C3AR1, CD83, CYBB*	[50,51,52]
Microbial killing (bacteria or yeast)	Microarray, RNAseq	*NR4A3, OLR1, TRAF1, CCRL2, PLAU, JMY, HS3STB1, LIF, IL1A, CXCL2, CREM, IRAK2, NR1D1, SFMBT2, SAMSN1, PNP, EIF2AK3, ZNF331, FRMD4B, TGIF1, VEGFA, EGR1, ILR1, C3AR1, C5AR1, TREM1, NLRP3, IRAK2, TICAM1, BFAR, **CXCL1**, BIRC3, **IL1B**, CXCL8, RIPK2, ADORA2A, GPR65, MAPK6, DUSP2, PTPRE, ICAM1, PLAUR, ACTG, TRIF, CD83, OSR1, TNF*	[51,53]
NETosis	Microarray	*GLA, RP11, ANXA1, EGR1, MAP4K5, SEMA7A, C3AR1, CYBB, H3F3C, IL18RAP, KLF2, **IL1B**, JUNB, MAP3K8, TNF, **CCL4**, **CXCL8***	[54,55]

**Table 2 cells-10-02406-t002:** Summary of neutrophil gene expression reported in major diseases. Genes featured are restricted to those highlighted by the original authors and/or the largest fold change reported by the study. Bold indicates common to the studies cited, underlined indicates common to multiple diseases. RNAseqRNA sequencing; NETosis—neutrophil extracellular trap formation.

Disease	Analysis Methods	Neutrophil Gene Expression Observed	Citations
Bacterial/Fungal infection	RNAseq	*Cxcl9, Gbp5, Ifit2, Cxcl11, Ifi205, **Il1a**, Ikbke, Csf3, Cxcl10, Ifnar1, Ifnar2, Tlr9, Il12a, Mx2, Ccl4, Il10, Nlrc5, Il1f6, Gbp3, Tnfrsf9, Phynin1, IL15ra, **IL6**, Ccl2, Tnf, Nos2, Cln3, Zfp64, Fundc2, Brix1, Slfn3, Yipf6, Dlg1, Smek1, Prr12, 2410016O06Rik, Ebna1bp2, Cdc37l1, Impa1, Gclc, Atp2c1, Las1l, Ahi1, Zfp192, Nt5e, Piga*	[69,70]
Viral Infection	RNAseq, scRNAseq	*IFNβ1, **ISG15**, IRF1, CXCL10, CASP4, NOD1, **PARP14**, EIF2AK2, XAF1, PARP9, PARP10, BCL6, Krt18, Prf1, IFITM1/3, RSAD2, CD274 (PD-L1), ZC3H12A, CD177, ARG1, S100A8, S100A9, CXCR4, SELL, SPI1*	[71,72,73]
Chronic Inflammation/Auto-immune	RNAseq	** *PAM* ** *, ADARB2, C5orf56, ICAM1, IL1B, CCR1, IFIH1, SOCS1, TNFAIP3, TNFSF13B, OAS1, OAS2, IFI35, IFI44, IFI44L, IFI6, IFIT2, IFIT3, IFIT5, IFITM1/3, APOBEC3B, AASS, ELF5, COL4A3, ZNF772, RNU5A-1, CEACAM19, OAS3, PGM5, TIAF1, LY6E, LILRB5, FAM21B, TBC1D15, HMGN3, PRKAR2B, PLXNB2, APMAP, PGM1, ACAP1, PYGL, S100A12, SRPK1, ACSL1, CLEC4D, MAP4K4, MAPK14, ACTB, PXK, TP53I11, ZDHHC3, BMX, PGD, SLC37A3, SLC26A8, ALOX5, KIF1B, PLP2, S100A9*	[74,75,76]
Cancer	Microarray, RNAseq, scRNAseq	*CD206, **CCL2**, ARG2, Nos2, CALR, LGMN, CXCL13, **CXCL10**, **TNF**, CASS4, CTSC, **CCL3**, **CD274 (PD-L1)**, TLR1, TLR4, IRAK2, IEX1, SOD2, GADD45b, BCL2A1, CD74, Cd1d-1, Psme1, H2-DMa, DMb1, Eb1, CCL17, CXCL9, CXCL16, CXCL1, CXCL2, IL1a, IL1b, IL12*	[77,78,79,80,81]
Cystic Fibrosis	RNAseq, scRNAseq	*FAM107A, SPOCK2, GZMH, GPRC5B, EDG1, TMEM35, A_24_P920715, AGT, ACVR1C, HBS1L, TRBV5-4, HBA2, PKN1, AEBP1, EPB41L2, GFAP, HBB, RNASEN, AK090499, HBA1, **CXCR4**, CD83, IRAK2, IRAK3, TRAF3, PLAU, CREM, HES4, KRAS, TRPM7, CCL3, CCL4, TLR2, IGF2R, KLRG1, CCR5, C4A, CCR4, CFD, SPP1, IFITM2, MSH2, FCGR2C, TREM1, CFD, CDKN1A, PLK3, ZC3HC1, MSH2, CYP26B1, KIF20B, CENPV, G0S2, CDK6, AHR, MYH6, PPP4R4, AIM2, NLRP3, S100A12*	[59,61,74,82]

## Data Availability

Not applicable.

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
