# Peer review of "Current Understanding of the Neutrophil Transcriptome in Health and Disease"

_cells, 2021, doi:10.3390/cells10092406_

Round 1

Reviewer 1 Report

The review article titled “Current understanding of the neutrophil transcriptome in health and disease” by Garratt is interesting and within scope of the journal cell. I have following comments to make the article reader friendly.

  1. The authors should incorporate tables which provide summary of diseases and key transcription molecules identified.

The authors should also make a table which shows how neutrophil transcription differs in healthy and disease state.

  1. How transcriptional studies will identify potential therapeutic targets? Please include in priority research section.
  2. How different tumors will regulate transcription of Neutrophils? Please include in cancer section
  3. Based on current transcriptional studies, can authors propose potential therapeutic molecules?
  4. Are there any clinical trials which have utilized neutrophil transcriptional studies?

Author Response

The review article titled “Current understanding of the neutrophil transcriptome in health and disease” by Garratt is interesting and within scope of the journal cell. I have following comments to make the article reader friendly.

Q1 The authors should incorporate tables which provide summary of diseases and key transcription molecules identified. The authors should also make a table which shows how neutrophil transcription differs in healthy and disease state.

R1 Thank you for the suggestions. I have created a table that summarises gene expression reported in neutrophil functions (Table 1) and gene expression in disease types (Table 2).

Q2 How transcriptional studies will identify potential therapeutic targets? Please include in priority research section.

R2: I feel this is beyond the scope of this review, so I have included two recent reviews for the reader to engage with if they would like more detail.

Q3How different tumors will regulate transcription of Neutrophils? Please include in cancer section

R3: Like the other sections, I have tried not to be too specific. I also do not operate in the cancer field, so I am fully qualified to deeply interpret the literature. But from my reading there has not been enough investigations in different tumours to form an accurate answer to this question.

Q4 Based on current transcriptional studies, can authors propose potential therapeutic molecules?

R4: In the interests of the paper length, I would prefer not to include this section. I have a study underway to collate publicly available data and work towards this goal, but we are still at the stage of data curation.

Q5 Are there any clinical trials which have utilized neutrophil transcriptional studies?

R5: To my knowledge there are none. There are a handful of trials appearing that are utilising neutrophil focused RNAseq as secondary or exploratory outcome measures. I have expanded this statement in the text.

Reviewer 2 Report

I have read the review entitled “Current understanding of the neutrophil transcriptome in health in disease” by Luke Garratt in great interest. The author gives a detailed and up-to-date account of the recent advance in our understanding of neutrophil transcriptional regulation in light of past perceptions. The review touches upon changes that relate to neutrophil developments, neutrophil structure, neutrophil transcriptional regulation in steady state and neutrophil transcriptional response in disease. In the section describing neutrophil transcription regulating the author goes into detail specifically in infection and inflammation and briefly discusses neutrophil transcription in cancer. This review is very informative, it is timely and it is very well written. I have only a few minor comments and I believe this manuscript should be accepted for publication once they are addressed.

  1. The subsection describing neutrophil transcriptional regulating in cancer is short and should be expanded. As the author states, there seem to be a continuum of transcriptional state in neutrophils in cancer and this needs some more detail.
  2. Some of the sentences are very long and difficult to follow – for example see paragraph 1 of the introduction.
  3. In the abstract – “Whilst initial efforts focused predominantly on understanding the extent biology….”. Not clear what “the extent biology” means.
  4. In line 104 “transcriptional” should be changed to “function”.

Author Response

Q1 The subsection describing neutrophil transcriptional regulating in cancer is short and should be expanded. As the author states, there seem to be a continuum of transcriptional state in neutrophils in cancer and this needs some more detail.

R1: I have endeavoured to expand this section as requested with three extra paragraphs.

Q2 Some of the sentences are very long and difficult to follow – for example see paragraph 1 of the introduction.

R2 I apologise for some sentence structures, the manuscript has now been fully proofed with this in mind.

Q3 In the abstract – “Whilst initial efforts focused predominantly on understanding the extent biology….”. Not clear what “the extent biology” means.

R3: Thank you to the reviewer for picking up on this error. The word “extent” was an wrongly autocorrected from “existing” and this line has now been amended to read “Whilst initial efforts focused predominantly on understanding existing biology, …”.  

Q4 In line 104 “transcriptional” should be changed to “function”.

R4: Corrected.